# The cryo-EM structure of a γ-TuSC elucidates architecture and regulation of minimal microtubule nucleation systems

Erik Zupa [1,2], Anjun Zheng[1,2], Annett Neuner[1], Martin Würtz [1], Peng Liu [1], Anna Böhler[1], Elmar Schiebel [1✉] & Stefan Pfeffer [1✉]

The nucleation of microtubules from αβ-tubulin subunits is mediated by γ-tubulin complexes, which vary in composition across organisms. Aiming to understand how de novo microtubule formation is achieved and regulated by a minimal microtubule nucleation system, we here determined the cryo-electron microscopy structure of the heterotetrameric γ-tubulin small complex (γ-TuSC) from *C. albicans* at near-atomic resolution. Compared to the vertebrate γ-tubulin ring complex (γ-TuRC), we observed a vastly remodeled interface between the SPC/GCP-γ-tubulin spokes, which stabilizes the complex and defines the γ-tubulin arrangement. The relative positioning of γ-tubulin subunits indicates that a conformational rearrangement of the complex is required for microtubule nucleation activity, which follows opposing directionality as predicted for the vertebrate γ-TuRC. Collectively, our data suggest that the assembly and regulation mechanisms of γ-tubulin complexes fundamentally differ between the microtubule nucleation systems in lower and higher eukaryotes.

[1] Zentrum für Molekulare Biologie der Universität Heidelberg, DKFZ-ZMBH Allianz, Im Neuenheimer Feld 282, 69120 Heidelberg, Germany. [2]These authors contributed equally: Erik Zupa, Anjun Zheng. ✉email: e.schiebel@zmbh.uni-heidelberg.de; s.pfeffer@zmbh.uni-heidelberg.de

Microtubules (MTs) have central functions in diverse cellular events, such as chromosome segregation, cytokinesis, and intracellular transport. For many of these functions, tightly regulated de novo formation of MTs from αβ-tubulin subunits is essential. While the composition of MT nucleation systems is to some extent heterogeneous across eukaryotic organisms, γ-tubulin, a member of the tubulin superfamily, is a universal determinant in this process[1,2].

γ-tubulin fulfils its function as part of a larger complex[3]. In *Saccharomyces cerevisiae*, Spindle Pole body Component 97 (Spc97) and Spc98 were identified to assemble with two molecules of yeast γ-tubulin (Tub4) into the 300-kDa heterotetrameric γ-tubulin small complex (γ-TuSC)[4,5], forming the core of the MT nucleation system in lower eukaryotes. By binding to the CM1 motif of the γ-TuSC-receptor protein Spc110, the γ-TuSC is anchored to the nuclear side of the yeast spindle pole body (SPB)[6] and concomitantly oligomerizes into a spiral-like assembly[7]. In a low-resolution cryo-electron microscopy (cryo-EM) reconstruction of the *S. cerevisiae* γ-TuSC oligomer, γ-tubulin molecules were observed to be arranged in a left-handed spiral with the same pitch and width as the αβ-tubulin subunits in cellular MTs[7,8]. Based on this observation, a model was proposed in which γ-tubulin complexes function as structural templates for MT nucleation. The potentially 14-spoked *S. cerevisiae* γ-TuSC spiral mimics one layer of the MT helix and serves as a starting point for nucleation of the newly forming MTs, thus overcoming the kinetic barrier of αβ-tubulin polymerization[7]. The minimal γ-TuSC/CM1 system present in *S. cerevisiae* is generally comparable to the essential building block of MT nucleation systems in many lower eukaryotes, including the yeast *Candida albicans*. However, it is to some extent unusual, because the *S. cerevisiae* genome does not encode for the protein Mozart1 (Mzt1), which is required for γ-TuSC function in most other organisms[9–12].

In higher eukaryotes, structural templates for MT nucleation are more versatile in terms of composition and regulation than in *S. cerevisiae* and *C. albicans*. In *Schizosaccharomyces pombe*, *Aspergillus nidulans*, and *Drosophila* species MTs are nucleated by two partially overlapping systems: the γ-TuSC/CM1 system and the large γ-tubulin ring complex (γ-TuRC), consisting of γ-tubulin, five structurally related orthologs of Spc97/98 (the "γ-tubulin complex proteins" GCP2 to GCP6) and further stably associated factors. In these organisms, the γ-TuSC/CM1 system is essential for viability while specific γ-TuRC components are dispensable[13,14]. Reasons for this flexibility and how it affects the structure of γ-tubulin complexes or the regulation of MT nucleation in these organisms are not understood.

In vertebrates, the γ-TuRC is an essential MT nucleator. Recently, three independent studies solved structures of the vertebrate γ-TuRC by cryo-EM single-particle analysis[15–17], establishing the overall molecular architecture of the complex. As anticipated, the γ-TuRC assumes the shape of an open left-handed spiral and consists of 14 heterodimeric spokes, each comprising one copy of γ-tubulin and one of five different GCP proteins. The spokes are partially interconnected by a structural scaffold that is located on the inside of the spiral and surprisingly contains one copy of actin. The arrangement of γ-tubulins in the isolated γ-TuRC does not completely reflect MT symmetry, indicating that conformational activation of the complex is required for MT nucleation[15–17].

While these cryo-EM reconstructions of the vertebrate γ-TuRC have provided fundamental mechanistic insights into MT nucleation in higher eukaryotes, comparable high-resolution structures of the γ-TuSC are still missing, limiting our understanding of molecular features specific to the γ-TuSC/CM1 systems and how this promotes γ-TuSC oligomerization and MT formation. Aiming to structurally characterize a typical minimal MT nucleation system, we here describe the cryo-EM structure of the γ-TuSC from *C. albicans* at near-atomic resolution and compare it to the vertebrate γ-TuSC unit as a prototypic representative of a high-complexity MT nucleation system. We observed a vastly remodeled interface between the two γ-tubulin-Spc97/98 spokes and opposing trends in γ-tubulin repositioning that are required for conformational activation. Both observations suggest that the assembly and regulation mechanisms fundamentally differ between the two MT nucleation systems.

## Results

**Overall architecture of the γ-TuSC from *C. albicans*.** We isolated recombinantly expressed *C. albicans* γ-TuSC from insect cells by affinity purification and anion-exchange chromatography (Supplementary Fig. 1) and used cryo-EM single particle analysis to obtain a structure at near-atomic resolution. In brief, γ-TuSC particles were located on cryo-EM micrographs and subjected to several consecutive rounds of computational particle sorting (Supplementary Fig. 2a, b). Refinement of the retained γ-TuSC particles yielded an initial cryo-EM density at 4.1 Å resolution (Supplementary Fig. 2c). Notably, local resolution of this reconstruction was highest at the interface between the two γ-tubulin spokes and decreased systematically towards the periphery of the complex (Supplementary Fig. 2c), suggesting conformational mobility of the two γ-TuSC spokes relative to each other. We thus divided the γ-TuSC into two individual segments, each comprising one Spc protein and one γ-tubulin molecule, and refined them separately. This approach increased global resolution to 3.6 Å (Supplementary Fig. 2d) and improved density quality towards the periphery of the complex. Consistently, local resolution increased overall and ranges from 3.3 to 4.7 Å (Supplementary Fig. 2d) in the final density (Fig. 1a). Next, we prepared homology models for *C. albicans* Spc97, Spc98 and γ-tubulin based on X-ray structures of human GCP4 and γ-tubulin, respectively. The homology models were adjusted and extended where required and refined against our cryo-EM density of the γ-TuSC (see Methods section). The final atomic model had good statistics and was validated against the cryo-EM reconstruction (Supplementary Fig. 2e and Table 1).

The γ-TuSC possesses a Y-like overall shape of 180 Å height and a width of 60-100 Å (Fig. 1b). It is composed of two interacting spokes, each comprising one copy of γ-tubulin and either one molecule of Spc97 or Spc98. Spc97/98 proteins contain two structurally conserved Gamma Ring Protein (GRIP) domains arranged into an elongated overall shape (Fig. 1c). The N-terminal GRIP1 domain mediates canonical interactions between the Spc subunits, while the C-terminal GRIP2 domain binds one copy of γ-tubulin. The mostly α-helical fold of the GRIP domains is conserved between Spc97 and Spc98. These α-helices are arranged into bundles that are stabilized by strong hydrophobic interactions (Supplementary Fig. 3a). Equally, the interface between the GRIP1 and GRIP2 domains contains hydrophobic patches that stabilize the overall domain arrangement (Supplementary Fig. 3b). For both Spc proteins, regions located N-terminal of the conserved GRIP1 domain could not be unambiguously traced in the cryo-EM reconstruction, indicating conformational flexibility of these regions (Fig. 1a, c) as already observed for homologous vertebrate GCPs in the context of the γ-TuRC. However, additional low-resolution density not covered by the core fold of Spc proteins is lining the exterior of the Spc97-Spc98 interface (Fig. 1a). Since we can exclude that proteins were co-purified at significant stoichiometry with the recombinant γ-TuSC (Supplementary Fig. 1b) and because this density loosely connects to the most N-terminal resolved Spc helices, it could

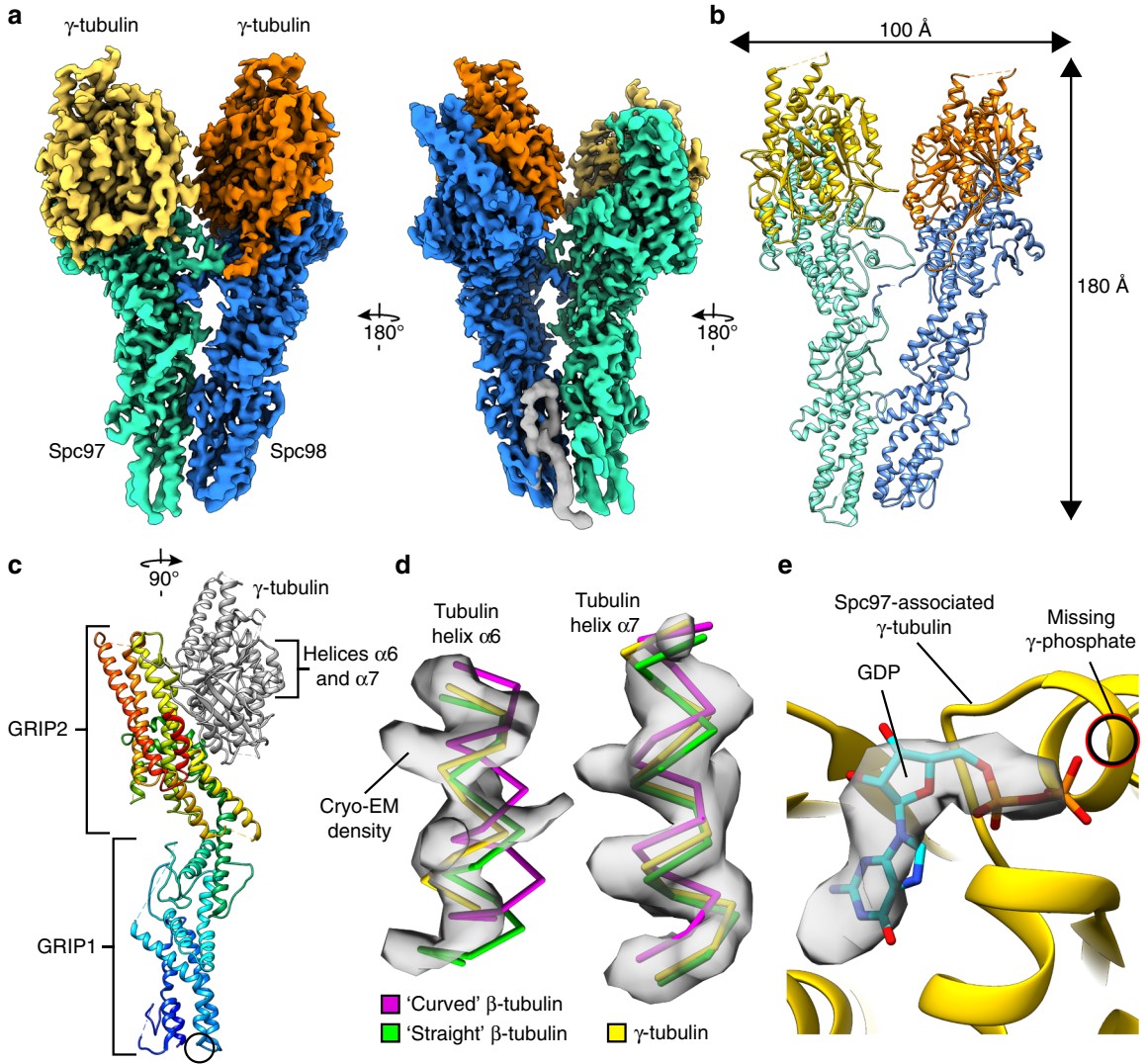

**Fig. 1 Molecular architecture of the γ-TuSC from *C. albicans*. a** Cryo-EM reconstruction of the γ-TuSC from *C. albicans*. Segmented cryo-EM densities for Spc97 (green), Spc98 (blue), and γ-tubulins (yellow, orange) are shown. Unassigned density is depicted in gray. The asterisk indicates the γ-tubulin nucleotide-binding pocket shown in panel **e**. **b** Atomic model of the γ-TuSC. Dimensions of the complex are given. Coloring as in **a**. **c** Atomic model of the Spc97-γ-tubulin heterodimer. The Spc97 model is colored from blue (N-terminus) to red (C-terminus). The location of the missing Spc97 N-terminus is indicated (black circle). The γ-tubulin model is shown in gray and α-helices α6 and α7 are annotated. **d** Helices α6 and α7 of γ-tubulin (yellow), 'straight' β-tubulin (green) and 'curved' β-tubulin (magenta) superposed to the cryo-EM density of Spc97-associated γ-tubulin. **e** Atomic model for GDP (cyan) superposed to cryo-EM density (gray) in the nucleotide-binding pocket of Spc97-associated γ-tubulin (yellow). The density does not indicate the presence of a γ-phosphate group. Zoom as indicated by the asterisk in panel **a**.

represent the missing N-terminal segments of either Spc97, Spc98, or both of the Spc proteins.

**γ-tubulin is in a MT nucleation-competent conformation**. For αβ-tubulin dimers, two different conformations have been described. According to the geometry of protofilaments that they occur in, these two conformations have been termed 'straight' or 'curved'[18]. To identify the conformation of γ-tubulins in our cryo-EM reconstruction, we compared them to X-ray structures of αβ-tubulin dimers in the two different conformations[19,20] (PDB 4FFB, PDB 5W3F), with a focus on the two α-helices in β-tubulin most variable between the two conformations. Visual inspection of overlap (Fig. 1d), cross-correlation between density segments (Table 2), and root mean square deviation (R.M.S.D) of protein backbone atoms in the atomic models (Table 2) consistently indicate that both copies of γ-tubulin in our cryo-EM reconstruction adopt a 'straight' conformation, as typically observed for αβ-tubulin dimers in stable and ordered microtubules[18]. Thus, γ-tubulins in the fungal γ-TuSC are in a conformation that can promote growth of 'straight' microtubule protofilaments without a conformational change[21]. Our cryo-EM density of the *C. albicans* γ-TuSC indicates binding of GDP, rather than GTP (Fig. 1e), and it remains to be addressed whether a nucleotide exchange to GTP is required to promote microtubule nucleation activity as suggested by analysis of γ-tubulin mutants in *S. cerevisiae*[21].

**Insertions in Spc97, Spc98, and γ-tubulin form an extended interface between γ-TuSC spokes**. Three intermolecular interfaces determine the quaternary structure of the γ-TuSC. Two of these interactions, namely the Spc97-γ-tubulin and the Spc98-γ-tubulin interactions, are required to form the two individual heterodimeric spokes. These two interfaces are mostly governed by electrostatic interactions between conserved charged patches

on both Spc proteins and γ-tubulin (Supplementary Fig. 3c, d). The third interface, namely between Spc97 and Spc98, assembles the two individual spokes into the heterotetrameric γ-TuSC. The Spc97-Spc98 interactions can be divided into two clusters. In *C. albicans*, the first cluster of interactions is mostly formed by hydrophobic interactions between the GRIP1 domains of Spc97 and Spc98 (Fig. 2a). While the positioning of this interaction area is conserved between *C. albicans* and vertebrates, the pre-dominantly hydrophobic character is not (Supplementary Fig. 4a). In contrast to the hydrophobic interactions in the

*C. albicans* γ-TuSC, this interface is mostly mediated by electrostatic interactions in the human γ-TuSC units (Fig. 2b and Supplementary Fig. 4b). The second cluster of interactions between Spc97 and Spc98 in the *C. albicans* γ-TuSC is formed by protein segments that are not conserved in the human homologs (Supplementary Fig. 5a–c). Generally, there are several short insertions in *C. albicans* Spc97 (Thr232-Asp272, Glu495-Pro502, Asn539-Ser566, and Ala686-Ile737) and Spc98 (Asn626-Leu656) that are missing in the vertebrate homologs and are partially resolved in our structure of the γ-TuSC (Fig. 2c, d and Supplementary Fig. 6). In particular, the long insertion in the Spc98 GRIP2 domain (Asn626-Leu656) is involved in extending the interface and bridges over the subunit interface to stably bind to the GRIP2 domain of Spc97 (Fig. 2c). The N-terminal region of this Spc98 insertion is resolved well from where it originates on Spc98 up to its binding site on Spc97. Similar as for the interface between the Spc97/98 GRIP1 domains, this interaction is stabilized by hydrophobic patches: Leu630, Leu631, Leu632, Phe636, Met637, and Leu639 of the Spc98 insertion are exposed to a hydrophobic binding pocket on the surface of Spc97 (Fig. 2d). The C-terminal segment of the Spc98 insertion is partially unordered and can only be traced back to Spc98 at low resolution. This long Spc98 insertion is stabilized by contacts with two additional insertions in the Spc97 GRIP2 domain (Val498-Tyr505) and the γ-tubulin N-terminal region (Thr38-Tyr72), respectively. The short Spc97 insertion clamps the N-terminal segment of the Spc98 insertion onto the Spc97 GRIP2 domain (Fig. 2c). The *C. albicans*-specific γ-tubulin insertion (Supplementary Fig. 5c) binds to the very C-terminal segment of the Spc98 insertion and thereby stabilizes it (Fig. 2e). Notably, this interaction is mediated by a composite hydrophobic groove formed by the C-terminal segment of the Spc98 insertion and the Spc98 GRIP2 domain into which a stretch of Iso-/Leucine residues of the γ-tubulin insertion binds (Fig. 2e). Such a hydrophobic groove is not present on Spc97 and consistently the γ-tubulin insertion is not resolved for the Spc97-γ-tubulin spoke. Collectively, insertions in three out of four γ-TuSC subunits (Spc97, Spc98, and Spc98-associated γ-tubulin) form a uniquely extended interface between the GRIP2 domains in the *C. albicans* γ-TuSC that is not present in the vertebrate γ-TuSC unit.

**The extended interface is central for γ-TuSC structure and function.** To explore the role of the extended γ-TuSC interface, we constructed deletion mutants missing the protein insertions in Spc98 (Spc98$^{\Delta D627\text{-}K650}$) and γ-tubulin (Tub4$^{\Delta T38\text{-}K71}$). Either one of the deletion mutants was recombinantly expressed in insect cells together with wild-type proteins of the other two γ-TuSC components from *C. albicans*. We isolated the mutant γ-TuSC (Supplementary Fig. 7a, b) following the same purification scheme as for the wild-type complex and structurally characterized the mutant and wild-type complexes by negative staining EM (Supplementary Fig. 8 and Fig. 3). The expression of mutant γ-TuSCs was overall lower as compared to the wild-type complex, already indicating that the missing protein insertions indeed destabilize the complex to some extent. First, we analyzed

**Table 1 Cryo-EM data collection, refinement and validation statistics.**

**#1 *Candida albicans* γ-TuSC (EMD-11835) (PDB-7ANZ)**

| Data collection and processing | Dataset 1 | Dataset 2 |
|---|---|---|
| Magnification | 81,000 | 81,000 |
| Voltage (kV) | 300 | 300 |
| Electron exposure (e–/Å$^2$) | 40 | 40 |
| Defocus range (μm) | −2 to −3 | −2 to −3 |
| Pixel size (Å) | 1.07 | 1.07 |
| Initial particle images (no.) | 309,744 | 1,502,064 |
| Final particle images (no.) | 45,351 | 228,975 |
| Merged set of particles | 274,326 | – |
| Map resolution (Å) | 3.6 | – |
| FSC threshold | 0.143 | |
| Model resolution range (Å) | 3.3-4.7 | – |
| Refinement | | |
| Initial model used (PDB code) | Homology models derived from human GCP4 (3RIP) and γ-tubulin (1Z5W) | – |
| Model resolution (Å) | 3.6    4.2 | – |
| FSC threshold | 0.143    0.5 | |
| Model resolution range (Å) | 3.5-4.3 | |
| Map sharpening *B* factor (Å$^2$) | −60.0 | |
| Model composition | | – |
| Non-hydrogen atoms | 16,759 | – |
| Protein residues | 2041 | – |
| Ligands | 0 | – |
| *B* factors (Å$^2$) | | – |
| Protein | 5.71/82.31/ 35.50 | – |
| Ligand | –/–/– | – |
| R.m.s. deviations | | – |
| Bond lengths (Å) | 0.003 | – |
| Bond angles (°) | 0.786 | – |
| Validation | | – |
| MolProbity score | 1.66 | – |
| Clashscore | 4.57 | – |
| Poor rotamers (%) | 0.31 | – |
| Ramachandran plot | | – |
| Favored (%) | 93.36 | – |
| Allowed (%) | 6.49 | – |
| Disallowed (%) | 0.15 | – |

**Table 2 Structural comparison between γ-tubulin and β-tubulin in two different conformations.**

| | R.M.S.D. of C$_\alpha$ backbone atoms | | Cross-correlation between density segments | |
|---|---|---|---|---|
| | β-tubulin curved | β-tubulin straight | β-tubulin curved | β-tubulin straight |
| γ-tubulin (Spc97) | 2.128 | 1.302 | 0.7876 | 0.8309 |
| γ-tubulin (Spc98) | 1.635 | 1.167 | 0.7950 | 0.8349 |

R.M.S.D. and cross-correlation values were computed for tubulin helices α6 and α7 after rigid body docking of complete β-tubulin models into the cryo-EM density.

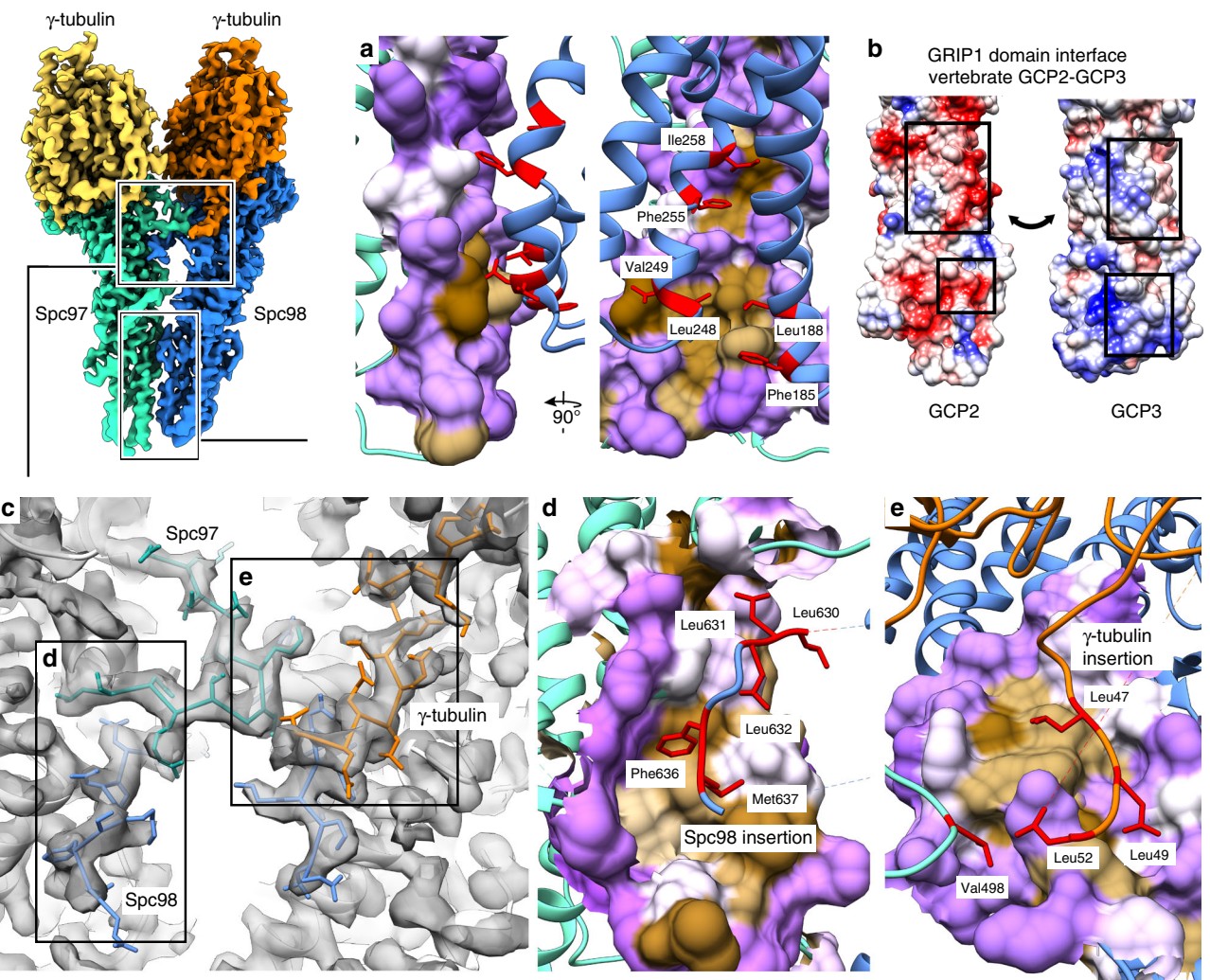

**Fig. 2 Protein insertions form an extended interface between γ-TuSC spokes. a** Zoom on the interface between the Spc97 and Spc98 GRIP1 domains as indicated in the overview. The Spc97 interface is represented as surface, color-coded according to hydrophobicity (brown: hydrophobic; purple: hydrophilic). Spc98 is represented as ribbon, with amino acid side chains mediating the hydrophobic interaction shown and depicted in red. **b** Interface between the human GCP2 (left) and GCP3 (left) GRIP1 domains. Complementary charged patches on the two surfaces mediate the interaction. Surface representation of the atomic model, colored according to charge (red: negative, blue: positive, and white: no charge). **c** Zoom on the fungi-specific extended interface between the γ-TuSC spokes as indicated in the overview. The atomic model (white) is shown superposed to the cryo-EM density (transparent). Fungi-specific loops mediating the interaction are colored as in Fig. 1a. **d** Zoom on the interface between Spc97 and the Spc98 insertion, as indicated in **c**. Representation as in **a**. **e** Zoom on the interface between Spc98 and the γ-tubulin insertion, as indicated in **c**. Representation as in **a**, but with Spc98 in surface representation.

the *SPC98* deletion mutant missing the Spc98 insertion Asp627-Lys650 (Spc98$^{\Delta D627-K650}$; Supplementary Figs. 7a and 8a). The general Y-like overall shape of the γ-TuSC was preserved, suggesting that the Spc97-Spc98 interaction via the two GRIP1 domains (Fig. 2a) was sufficient to form a stable complex. However, inspection of representative particles and 2D class averages showed that the angle between the GRIP2 domains of Spc97 and mutant Spc98 was much more variable and significantly wider ('straddled') as for the wild-type complex in 51% of γ-TuSC particles (Fig. 3b). Next, we analyzed the *TUB4* deletion mutant missing the γ-tubulin insertion Thr38-Lys71 (Tub4$^{\Delta T38-K71}$). Similar to the Spc98$^{\Delta D627-K650}$ mutant, we observed 2D classes corresponding to complexes with regular and 'straddled' appearance (Fig. 3c). However, the proportion of particles with a 'straddled' appearance was significantly lower (28%) than in Spc98$^{\Delta D627-K650}$ mutant, suggesting that the γ-tubulin insertion has only a secondary function in stabilizing the complex. Collectively, these data suggest that the extended

interface built around the Spc98 insertion is centrally involved in defining the overall shape of the *C. albicans* γ-TuSC by aligning the GRIP2 domains and their associated γ-tubulins, which can be expected to be of high importance for MT nucleation activity of the γ-TuSC.

To test whether the mutant γ-TuSC variants maintain microtubule nucleation activity, we turned to a genetically better accessible model system, the yeast *S. cerevisiae*. While the Spc97 and Tub4 insertions are present only in *C. albicans*, the Spc98 insertion is present in both *C. albicans* (D627-K650) and *S. cerevisiae* (K674-H713) and only slightly varies in its length between the two species of fungi (Supplementary Fig. 5a–c). This suggests a similar function in both organisms, which also could be confirmed from a structural perspective by a preliminary cryo-EM study on the *S. cerevisiae* γ-TuSC[22]. We thus performed a plasmid shuffle experiment using *S. cerevisiae* spc98Δ pRS316-*SPC98* cells (Fig. 3d and Supplementary Fig. 9a) expressing either *LEU2*-based plasmid-encoded *S. cerevisiae* SPC98 wild-type or

spc98^{ΔK674-H713} mutant genes. Cells with *LEU2-SPC98* grew at all temperatures on 5-FOA plates, which select for cells that spontaneously lost the plasmid pRS315-*SPC98*. While *LEU2-spc98^{ΔK674-H713}* could partially restore cell growth at 23 °C and 30 °C, cells were strongly impaired for growth at 37 °C indicating a conditional lethal growth defect. This suggests that the

Spc98^{ΔK674-H713} mutant γ-TuSC maintains residual MT nucleation activity, but is not as efficient as the wild-type complex. Consistently, *S. cerevisiae spc98^{ΔK674-H713}* cells expressing α-tubulin Tub1 fused to yeast enhanced GFP (Tub1-yeGFP) and the SPB marker Spc42-mCherry showed defective mitotic spindles after 3 h incubation at 37 °C (Supplementary Fig. 9b). Mitotic

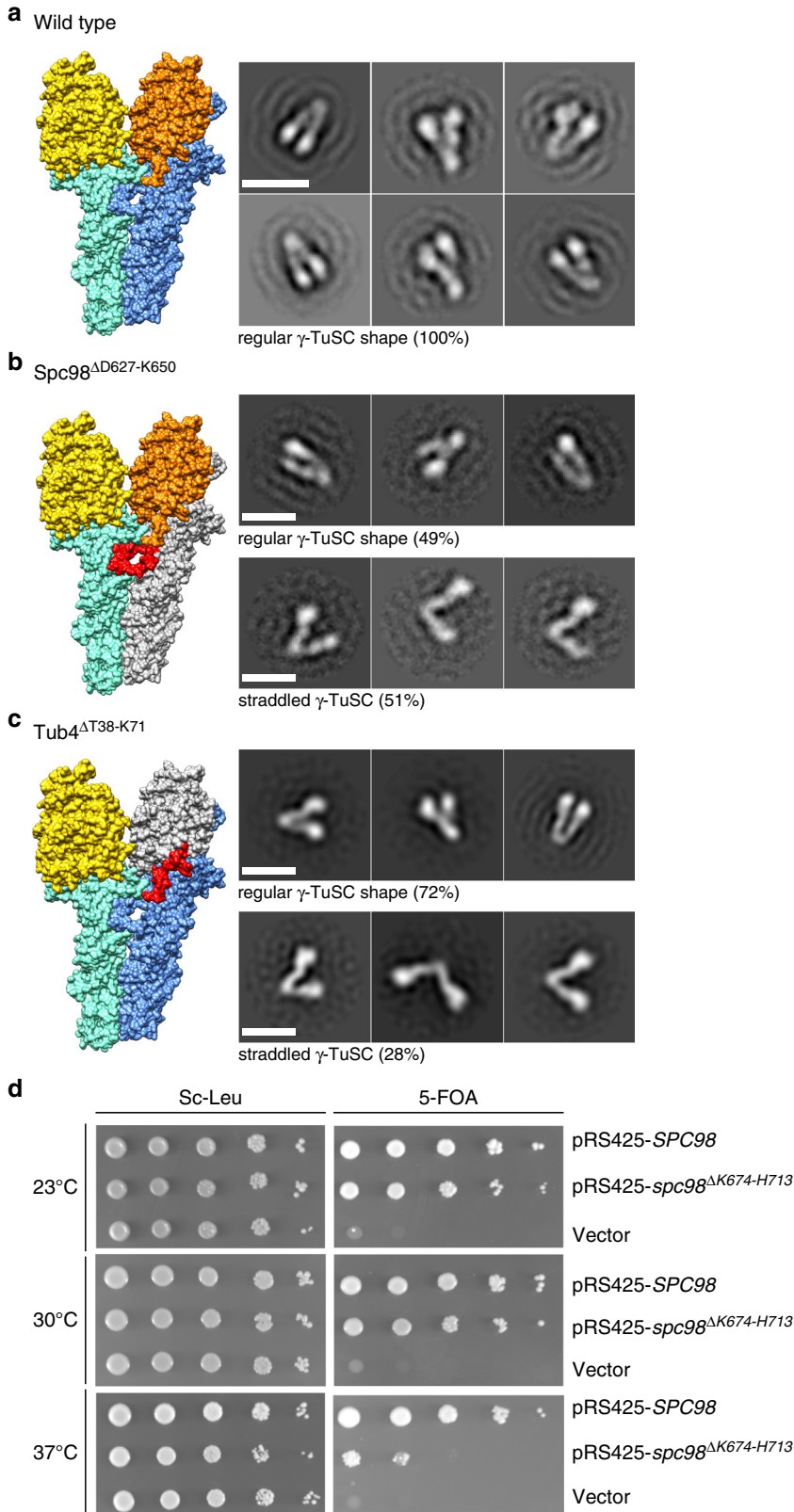

**Fig. 3 The extended interface is central for γ-TuSC function and stability in *C. albicans*. a** Wild-type γ-TuSC were analyzed by negative stain EM. Representative 2D classes are shown. 77,858 particles were included in the final set of 2D classes. Exclusively γ-TuSC particles with regular conformation are observed. Scale bar corresponds to 20 nm. **b, c** Spc98$^{ΔD627-K650}$ mutant γ-TuSC (**b**) or Tub4$^{ΔT38-K71}$ mutant γ-TuSC (**c**) were analyzed by negative stain EM. Representative 2D classes for the different particle populations are shown. Relative abundance is given. The absolute number of particles used for computing the ratios were 11901 for the Spc98$^{ΔD627-K650}$ mutant γ-TuSC and 15450 for the Tub4$^{ΔT38-K71}$ mutant γ-TuSC. Scale bar corresponds to 20 nm. Mutated subunits of the γ-TuSC are shown in gray and the deleted insertions are shown in red. Only 2D classes depicting a clearly discernable V-shape were retained for the analysis. Data for each experiment were acquired in one imaging session. **d** Viability of *S. cerevisiae* spc98$^{ΔK674-H713}$ cells. Indicated plasmid (pRS425) encoded alleles were transformed into *S. cerevisiae* strain ESM243-1 (*MATa Δspc98::HIS3* pRS316-*SPC98*). Serial dilutions of transformants were tested for growth at the indicated temperatures on SC-Leu and 5-FOA plates for 2–3 days. All cells grew equally on SC-Leu plates because of the presence of pRS316-*SPC98*. 5-FOA only allows growth of cells that spontaneously lost the *URA3*-based plasmid pRS316-*SPC98*. Cells with the empty pRS425 plasmid did not grow on 5-FOA at all temperatures because *SPC98* is an essential gene and therefore loss of pRS316-*SPC98* is lethal. The conditional lethal phenotype of spc98$^{ΔK674-H713}$ on pRS425 became apparent at 37 °C. The experiment was repeated three times with the same outcome.

spc98$^{ΔK674-H713}$ cells contained monopolar, broken or multi-polar spindles and cytoplasmic microtubules were frequently overly long. The *SPC98* control cells incubated at 37 °C assembled proper bipolar spindles (Supplementary Fig. 9c). Thus, deletion of the insertion in *SPC98* affects the ability of the γ-TuSC to organize a functional set of microtubules.

**The Spc98 insertion may have an evolutionarily compensatory role.** Having observed the important role of the Spc98 insertion in stabilizing the *C. albicans* γ-TuSC, we sought to identify structural features that could take over a similar function in organisms lacking the Spc98/GCP3 insertion. For this purpose, we systematically compared the interactions between Spc/GCP-γ-tubulin heterodimers in our model of the *C. albicans* γ-TuSC and the human γ-TuSC unit. As anticipated in the absence of the Spc98 insertion, direct interaction between human GCP2 and GCP3 is limited to the GRIP1 domains. However, the interface between the two spokes is extended by direct electrostatic interactions between complementary charged patches on the γ-tubulin molecules, which are in direct proximity in the human γ-TuSC unit (Supplementary Fig. 10a).

To explore whether such electrostatic interactions between neighboring γ-tubulin molecules might be a general feature of the γ-TuSC unit in higher eukaryotes, we selected a wide range of organisms and systematically compared the sequences for Spc98/GCP3 and γ-tubulin. Indeed, the key residues mediating the electrostatic interactions between γ-tubulin molecules are widely conserved, except for organisms that possess the Spc98/GCP3 insertion (Supplementary Fig. 10b, c). This pattern indicates that the electrostatic interaction between γ-tubulins of neighboring spokes provides a similar stabilizing effect as the Spc98/GCP3 insertion in *C. albicans* and therefore might be able to replace the Spc98/GCP3 insertion functionally.

Another striking difference between the two groups of organisms that either possess or lack the Spc98/GCP3 insertion is the complexity of their MT nucleation system. Within the range of organisms that we analyzed, the genome of almost all organisms lacking the Spc98/GCP3 insertion encodes for all five GCP proteins known to be required for the formation of the large γ-TuRC (Supplementary Fig. 11 and Supplementary Tables 1 and 2), suggesting that γ-TuSC units are mostly present in the context of the pre-assembled γ-TuRC in these organisms. Thus, in addition to the stabilizing effect of the γ-tubulin interaction, the γ-TuSC-like units could be further structurally supported in these organisms by additional lateral interactions with neighboring spokes in the γ-TuRC. The presence of the full γ-TuRC system in evolutionary lineages that have emerged long before fungi during evolution further indicates that the γ-TuSC/CM1 system as observed in *C. albicans* and *S. cerevisiae* emerged

**Table 3 R.M.S.D. of Cα backbone atoms between γ-TuSC segments in different overall conformations.**

|  | *Homo sapiens* to *S. cerevisiae* in 'closed' state | *C. albicans* to *S. cerevisiae* in 'closed' state | *C. albicans* to *S. cerevisiae* in 'open' state |
|---|---|---|---|
| Spc97 GRIP1 | 4.088 | 3.262 | 2.661 |
| Spc97 GRIP2 | 2.447 | 1.483 | 2.716 |
| Spc98 GRIP1 | 7.310 | 5.579 | 5.872 |
| Spc98 GRIP2 | 7.784 | 13.692 | 7.234 |
| Spc97 γ-tubulin | 2.934 | 1.432 | 3.728 |
| Spc98 γ-tubulin | 7.199 | 18.591 | 4.490 |

by compositional simplification from the γ-TuRC in order to form a specialized MT nucleation system in yeast, or more specifically in *Saccharomycetes* (Supplementary Fig. 11). This suggests that the stabilizing role of the Spc98 insertion is evolutionarily compensating for the loss of an extended interaction network that γ-TuSC units may experience in the context of the pre-assembled γ-TuRC in many other organisms.

**The *C. albicans* γ-TuSC requires conformational rearrangement during or after oligomerization to serve as a structural template for microtubule nucleation.** Available cryo-EM structures of γ-tubulin complexes have established that the relative arrangement of spokes and even the relative arrangement of GRIP domains within individual spokes can be variable. These variations were linked to structural differences between Spc/GCP paralogues (e.g. GCP2-6 in vertebrate γ-TuRC[15,16]), but also to conformational plasticity within the individual spokes that might relate to different functional contexts[8]. During MT nucleation, γ-tubulin complexes are thought to act as structural templates reflecting the MT symmetry and thus the relative arrangement of Spc/GCP proteins and their associated γ-tubulins within these complexes is of high importance for their function. We therefore systematically compared the relative arrangement of Spc97/98 proteins and γ-tubulins between our cryo-EM reconstruction of the *C. albicans* γ-TuSC, recent high-resolution structures of the vertebrate γ-TuRC and previous low-resolution cryo-EM structures of *S. cerevisiae* γ-TuSC helical oligomers[7,15,16]. For this comparison, structures of the γ-TuSC units were superposed according to the Spc97/GCP2 spoke and the displacement of GRIP domains and γ-tubulins was quantified by R.M.S.D. of their protein backbone atoms (Table 3). For selected pairs, we visualized the underlying rearrangements by displaying the trajectories linking the corresponding protein backbone atoms between the two conformations.

First, we compared our structure of the *C. albicans* γ-TuSC to the *S. cerevisiae* γ-TuSC as observed in the 'open' γ-TuSC

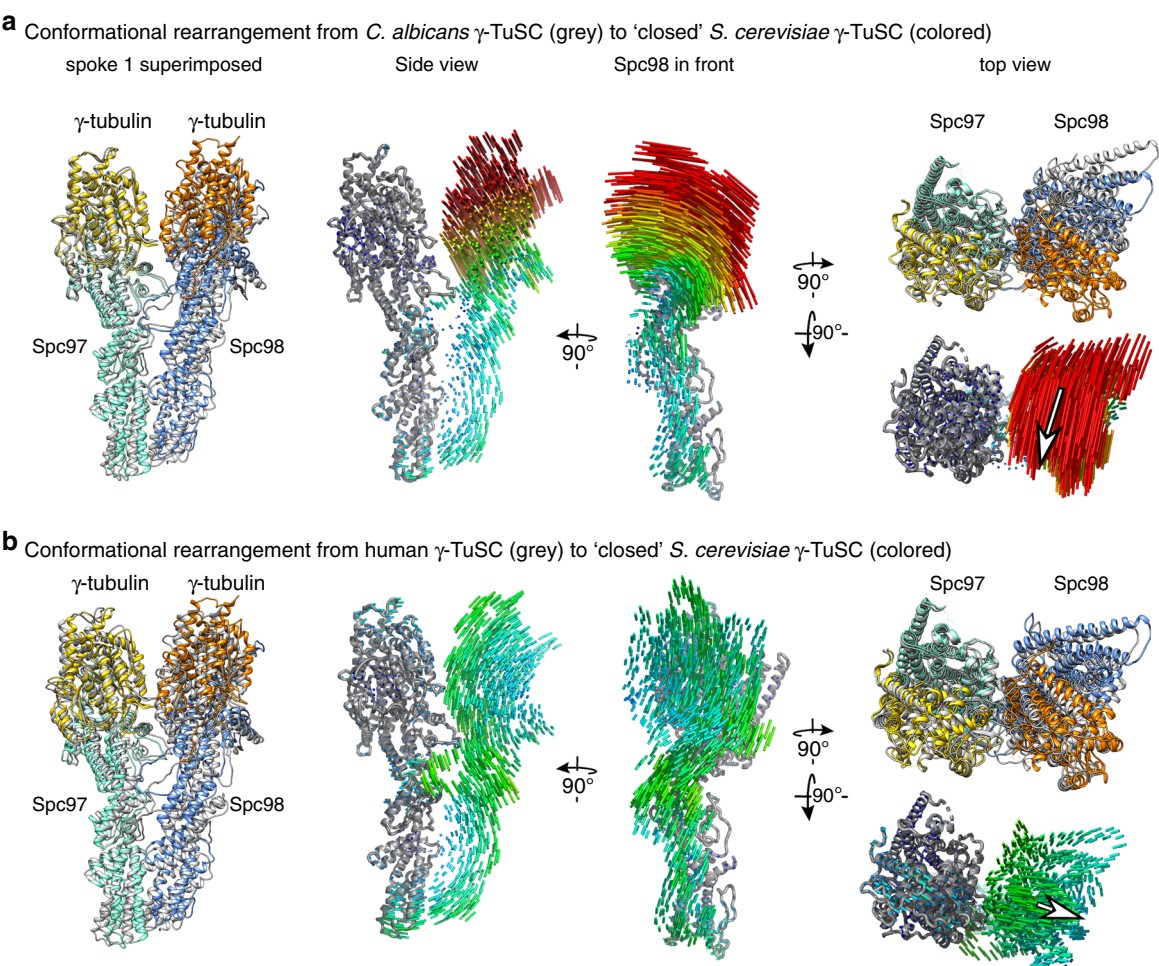

**Fig. 4 The *C. albicans* γ-TuSC requires conformational rearrangements during or after oligomerization.** Atomic models for the *C. albicans* (**a**, gray) and vertebrate (**b**, gray) γ-TuSC units were superposed to the *S. cerevisiae* γ-TuSC in the closed conformation (colored) according to the first spoke (Spc97/GCP2 plus γ-tubulin). The displacement of domains was visualized by trajectories linking the backbone atoms in the two superposed models. For orientation, atomic models for the first spoke are superposed to trajectories. Trajectories are color-coded from low (blue) to high (red) motion. In the 'top view', arrows representing the average direction and magnitude of γ-tubulin displacement are superposed to the trajectories.

oligomer helix, suggested to be inactive in MT nucleation, because the γ-tubulin positioning does not recapitulate the helical parameters of a MT. We observed high similarity between the two complexes with no major domain rearrangements and overall low R.M.S.D. values for all Spc97/98 domains and γ-tubulins (Table 3). Next, we compared our structure of the *C. albicans* γ-TuSC to the *S. cerevisiae* γ-TuSC as observed in the 'closed' γ-TuSC oligomer helix, which accurately recapitulates the helical parameters of a MT[7,8]. While the GRIP1 domains are arranged similarly, the GRIP2 domain of Spc98 and the associated γ-tubulin molecule are both strongly displaced, suggesting that a conformational rearrangement of GRIP domains within the spoke is required for assembly of an active MT nucleation template (Fig. 4a and Table 3). This movement can be approximated by a tilting motion towards the helical axis of the γ-TuSC oligomer, moving the centers of the two γ-tubulin molecules 5 Å closer to each other. This indicates that the *C. albicans* γ-TuSC needs to undergo a profound conformational rearrangement during or after oligomerization to act as a structural template for microtubule nucleation. Notably, this conformational rearrangement would require adjustments in the extended interface between Spc97 and Spc98 (Fig. 2), as the GRIP2 domains of both Spc proteins would come significantly closer.

Lastly, we compared the vertebrate γ-TuSC unit as observed in the 14-spoked human γ-TuRC helix to the *S. cerevisiae* γ-TuSC oligomer in the 'closed' conformation (Fig. 4b and Table 3), observing an overall similar conformation between the two complexes. In particular, the relative arrangement of γ-tubulin molecules in the two complexes is more similar as in case of the *C. albicans* γ-TuSC. Nevertheless, the γ-tubulin molecules in the vertebrate γ-TuSC unit still have to move 2.5 Å further apart to perfectly reflect MT symmetry. Thus, while the low nucleation activity of the isolated γ-TuRC has mostly been attributed to its asymmetric shape originating from the incorporation of GCP4-6 into the complex[15–17], clearly also the GCP2-GCP3 γ-TuSC units have to undergo a rearrangement.

## Discussion

Our cryo-EM reconstruction of the *C. albicans* γ-TuSC provides high-resolution structural insights into the essential building block of the γ-TuSC/CM1 MT nucleation system. Our structure thus complements the recent high-resolution cryo-EM reconstructions of the compositionally more elaborate vertebrate γ-TuRC system and allows for a detailed comparison of molecular features relevant for γ-tubulin complex assembly and regulation in the two different MT nucleation systems. Notably, many aspects observed for the γ-TuSC/CM1 system in *C. albicans* seem

to be generally recapitulated in the related fungus *S. cerevisiae*, as suggested by a preliminary cryo-EM study[22], but structural data for this study are not publicly available yet, precluding any further in-depth comparison.

The overall structures of the *C. albicans* and vertebrate γ-TuSC units are comparable, but the interface between the two spokes has strongly diverged. The interaction between the GRIP1 domains of Spc97/GCP2 and Spc98/GCP3 has shifted from a mostly electrostatic character in vertebrates towards a more hydrophobic character in *C. albicans*. Most surprisingly, the interface between Spc97/GCP2 and Spc98/GCP3 has been extended in *C. albicans* by insertions in all γ-TuSC components that additionally link and align the GRIP2 domains of both spokes. Conversely, the *C. albicans* γ-TuSC lacks the direct interactions between adjacent γ-tubulin molecules that likely fulfil a similar function in the vertebrate γ-TuSC unit. This suggests two different solutions to stabilizing the elongated overall structure of the γ-TuSC.

Both in vertebrates and *C. albicans*, the relative spatial arrangement of γ-tubulin molecules in the γ-TuSC unit differs from MT symmetry, suggesting that both complexes represent inactive conformations. However, the magnitude and directionality of rearrangements that have to occur to transition the complexes into a MT nucleation-competent conformation are following opposing trends. In *C. albicans*, the γ-tubulin arrangement strongly differs from MT symmetry, requiring γ-tubulin molecules to move 5 Å towards each other during activation (Fig. 4a). In contrast, the γ-tubulin arrangement in the vertebrate γ-TuSC unit already more closely resembles MT symmetry and the γ-tubulin molecules have to move only slightly apart by 2.5 Å. These opposing trends in γ-tubulin repositioning during γ-tubulin complex activation indicate that different conformational switches might be involved in the two systems. Notably, it cannot be entirely excluded that the opposing trends in predicted γ-tubulin repositioning are linked to the different assembly states (individual *C. albicans* γ-TuSC vs. human γ-TuSC in the context of the γ-TuRC spiral). However, a similar conformation as observed for the isolated *C. albicans* γ-TuSC in our study can also be found for the oligomeric *S. cerevisiae* γ-TuSC[7] (Table 3), arguing against a direct effect of the assembly state on the overall conformation of the γ-TuSC.

One central feature that could be underlying differing conformational switches is the strongly diverged interface between the SPC/GCP-γ-tubulin spokes in the two systems. In particular, our functional and structural analysis of wild-type and mutant *C. albicans* γ-TuSC indicates a central function of the Spc98 insertion in determining the overall conformation of the fungal complex and thus the γ-tubulin orientation. This underlines the potential of the Spc98 insertion to conformationally regulate the nucleation activity by controlling the γ-tubulin arrangement. Such a mechanism of activity regulation would be strictly limited to organisms possessing the Spc98 insertion and therefore supports the idea of a different mode of regulation as compared to the vertebrate γ-TuRC.

For both MT nucleation systems, conformational activation of nucleation activity has been proposed to depend on additional factors binding to the γ-tubulin complexes. For the *C. albicans* γ-TuSC, we observed factor-independent propensity for conformational plasticity during cryo-EM data processing (Supplementary Fig. 2c, d), but the range of flexibility was small in comparison to the conformational changes required for activation. More large-scale conformational rearrangements of the γ-TuSC could therefore occur (i) concomitantly with oligomerization of the γ-TuSC, triggered by additional interactions with the spokes of neighboring γ-TuSC units, (ii) they could be induced by phosphorylation of γ-TuSC components as previously observed for *S. cerevisiae*[23–26], (iii) or they could indeed be triggered by regulatory factors after oligomerization, as it was suggested for binding of Spc110 based on low-resolution cryo-EM structures of *S. cerevisiae* γ-TuSC oligomers[7]. The latter situation would be reminiscent of the vertebrate γ-TuRC, which is pre-assembled in a MT nucleation incompetent state and requires the action of the CM1-motif containing protein CDK5RAP2 and additional unknown factors to be conformationally activated[15].

In most organisms, including humans and *C. albicans*, the small protein Mzt1 has been proposed to be involved in γ-TuSC oligomerization by binding to the N-terminus of Spc98[9,27]. Interestingly, our cryo-EM reconstruction suggests that the segments N-terminal of the GRIP1 domain in either Spc97, Spc98, or both Spc proteins have a role in shaping the interface between the Spc proteins (Fig. 1a). Thus, binding of Mzt1 to the N-terminus of Spc98 could position it optimally to manipulate the critical oligomerization interface between γ-TuSC units, either by contributing to the interface itself, or by modulating the conformation of the Spc98 N-terminus and consequently the Spc97-Spc98 interface. A comparable positioning of GCP N-terminal segments was suggested for the GCP2-GCP3 interface in the vertebrate γ-TuSC units[16], indicating a similar situation in both MT nucleation systems in this respect.

Finally, our study provides an alternative perspective on the evolutionary relationship between the two MT nucleation systems. It seems conceivable that the elaborate γ-TuRC system could have emerged from the γ-TuSC/CM1 system by duplication of the genes coding for Spc97/98 (GCP2/3). The existence of organisms in which both systems are present, e.g. *S. pombe* and *Drosophila* species, may support this idea. In these organisms, the γ-TuSC/CM1 system is essential and sufficient for MT nucleation, but the accessory functions provided by the γ-TuRC components promote nucleation activity. These organisms thus might reflect a transition phase between the two systems, before, in higher eukaryotes including vertebrates, the γ-TuRC becomes the primary MT nucleator and the γ-TuSC system disappears. However, there is also indication for an inverse evolutionary relationship between the γ-TuRC and the γ-TuSC/CM1 systems. Our systematic analysis of Spc/GCP primary sequences and the composition of MT nucleation systems across a wide range of organisms indicates that the γ-TuSC/CM1 system found in *Saccharomycetes* (including *C. albicans* and *S. cerevisiae*) emerged by compositional simplification from the γ-TuRC to form a specialized MT nucleation system in yeast. For *S. cerevisiae*, such a reduction of complexity in the MT nucleation system could be well explained as an adaptation to the comparably simple architecture and regulation of the MT system. Compared to most other organisms, the absolute number of MTs is rather low, all of them are nucleated from the spindle pole body and all MTs are probably nucleated at the same time during spindle pole body duplication. The reduced spatial and temporal complexity of MT nucleation in yeast might thus not require the high degree of flexibility and versatility the more complex γ-TuRC system provides. At the same time, a simplified MT nucleation system could also provide a competitive advantage in an environment of limited resources by allowing faster cell division and thus duplication times.

Collectively, this systematic structural comparison of prototypic γ-tubulin complexes from two different MT nucleation systems reveals many similarities in terms of overall molecular architecture, but it also indicates surprising discrepancies between the systems. Further studies are required to elucidate how these structural differences impact in detail on the oligomerization of γ-tubulin complexes and the regulation of the MT nucleation reaction.

## Methods

**Plasmid construction**. Synthesized codon-optimized coding regions (G blocks, Integrated DNA Technologies) of *Candida albicans* SPC97 (orf19.708), SPC98 (orf19.2600), and γ-tubulin (orf19.1238) were subcloned into the insect cell expression plasmid pFastBac[9]. The spc98[ΔD627-K650] and tub4[ΔT38-K71] deletions in pFastBac were constructed by PCR. The primers are listed in Supplementary Table 3. All DNA constructs were confirmed by sequencing.

**Purification of wild-type and mutant *C. albicans* γ-TuSCs**. Protein expression in SF21 insect cells using the Bac-to-Bac® Baculovirus Expression System (Invitrogen life technologies) and subsequent purification of wild-type and mutant γ-TuSC complexes was performed according to an established protocol[9] and as described in the following paragraph. Briefly, pFastBac plasmids containing either *C. albicans* wild-type Spc97[9], wild-type Spc98[9], wild-type Tub4[9], mutant Spc98[ΔD627-K650], or mutant Tub4[ΔT38-K71] were sequenced and transformed into the DH10Bac™ competent cells (obtained from Prof. Imre Berger, Grenoble) to isolate the recombinant bacmid DNA by selecting the white colonies on plates containing X-gal, IPTG, and antibiotics (kanamycin, gentamicin, and tetracycline). After the bacmid DNA was purified, ~2–3 µg of bacmid DNA was transfected into SF21 cells (obtained from the European Molecular Biology Laboratory Protein Expression and Purification Facility) and P1 baculovirus was harvested 72 h post-transfection after incubation at 27 °C incubator. Then 2 ml of baculovirus-containing supernatant was mixed with 50 ml insect cell culture at a concentration of $1 \times 10^6$ cells/ml. The cells were cultured for 5 days to harvest the P2 baculovirus, which was stored at −80 °C. For protein purification, P2 viruses for the different γ-TuSC components were mixed as required and cells were harvested 48 h after infection. Cells were lysed in RSB buffer (10 mM NaCl, 1.5 mM $MgCl_2$, 10 mM Tris pH 7.4, 1 mM DTT, 250 µM EGTA, and one Roche inhibitor tablet/50 ml). After centrifugation of the lysate ($179,000 \times g$, 35 min, SORVALL Discovery 90SE ultracentrifuge, 50.2 Ti rotor), the supernatant was incubated with Protino® Ni-TED Resin (MACHEREY-NAGEL) for 1 h at 4 °C, and protein was eluted with 500 mM Imidazole. The eluted protein was loaded onto a Mono Q® 5/50 GL column (GE Healthcare) and further purified by anion-exchange chromatography using buffer A (20 mM NaCl, 50 mM Tris pH 7.4, and 0.5 mM EGTA) and buffer B (1 M NaCl, 50 mM Tris pH 7.4, and 0.5 mM EGTA). The protein peaks were verified by SDS-PAGE and used for negative stain EM. Uncropped raw images of the SDS-PAGE gels are included in the Source Data. All anion-exchange chromatography data were plotted using Prism 6.1 (GraphPad Software).

**Plasmid shuffle experiment**. The *S. cerevisiae* spc98[ΔK674-H713] deletion was constructed by PCR. The primers are listed in Supplementary Table 3. The pRS425 vector was double digested (ApaI, SacI) and either *S. cerevisiae* wild-type SPC98 or *S. cerevisiae* mutant spc98[ΔK674-H713] were integrated via recombination. The plasmids were transformed into the *S. cerevisiae* ESM243-1 strain (MATa Δspc98::HIS3 pRS316-SPC98). The transformed cells were spotted on SC-Leu and 5-FOA plates separately at 23 °C, 30 °C and 37 °C and incubated for 2–3 days. Images were obtained on a LAS-4000 imaging system (Fujifilm Life Science) with an exposure time of 1/15 s and Epi-illumination.

***S. cerevisiae* spc98[ΔK674-H713] phenotype analysis**. To analyze the phenotype of *S. cerevisiae* spc98[ΔK674-H713] in vivo, the pRS425-spc98[ΔK674-H713] plasmid was transformed into *S. cerevisiae* ESM243-2 strain (MATa ura3-52 lys2-801 ade2-101:: pRS402-yeGFP-TUB1 trp1Δ63 his3Δ200 leu2Δ1 Δspc98::HIS3 pRS316-SPC98 SPC42-mCherry-natNT2). Cells were grown on 5-FOA and then cultured in SC-Leu medium at 23 °C overnight and the $OD_{600}$ value was calibrated to 0.5–0.8. Subsequently, cells were continuously cultured at 23 °C and 37 °C separately for 3 h. In all, 5 µl of cells were dropped onto glass slides and then covered with a 35-mm glass dish. Images were acquired with a DeltaVision RT system (Applied Precision) on an Olympus IX71 microscope equipped with 100X NA UPlanSAPO objective lens (Olympus), and TRITC and FITC channels were selected with the softWoRx software (Applied Precision) to observe different fluorescence proteins. To compare fluorescence intensities, all quantification experiments were conducted at the same exposure and illumination settings, the ×100 objective, and a 2 × 2 binning. Image processing and analysis was performed semiautomated with the open-source software package ImageJ 1.46r (National Institutes of Health).

**Cryo-EM sample preparation and data acquisition**. Quantifoil holey carbon grids (Cu R2/1; 300 mesh) were glow discharged in a Gatan Solarus 950 plasma cleaner for 20 s. In all, 4.0 µl of purified *C. albicans* γ-TuSC were applied on the grid and incubated for 30 s. The grids were blotted for 5–10 s and subsequently plunge frozen into liquid ethane cooled by liquid nitrogen using a Vitrobot Mark IV (Thermo Fisher Scientific). Cryo-EM data were acquired in two sessions on a Titan Krios transmission electron microscope (Thermo Fisher Scientific) operated at 300 kV and equipped with an energy-filtered Gatan K2 (Gatan, Inc.) direct detector operated in dose fractionation mode (20 frames/frame stack). Data were collected at a pixel size of 1.07 Å with a cumulative dose of ~40 e/Å² using SerialEM[28]. Defocus and eucentric height were adjusted for each pre-selected hole automatically. Data were collected at a defocus range from −2 to −3 µm with

4 frame stacks per hole. Data were acquired in two imaging sessions on two different EM grids.

**Cryo-EM data processing**. Image processing was performed in Relion 3.0 Beta[29], if not stated otherwise. Frame stacks from both sessions were motion corrected using MotionCor2.0[30] with 5 × 5 patches. The contrast transfer function (CTF) of motion-corrected micrographs was estimated using gCTF[31]. Due to the high particle density on the micrographs, initial particle selection was based on a regular grid with a spacing of 100 pixels (corresponding to 10.7 nm). Particles were extracted at a pixel size of 4.28 Å and a box size of 64 pixels. Particles were grouped into four subsets and each subset was individually sorted by an initial round of 3D classification. Translational sampling was selected to cover 20 pixels (corresponding to 8.6 nm) at an increment of 2 pixels, which allowed to search the entire area between grid points. Particles were sorted into six classes, with an initial low-pass filter of 15 Å, a T-factor of 10 and a shape mask. As initial reference, we used a cryo-EM density of the γ-TuSC segmented from the γ-TuSC oligomer in the closed conformation (EMD-2799). Each subset of particles produced one to two classes with high-resolution features. Particles contained in these classes were selected and merged and duplicate particles originating from the large translational search range were excluded.

The retained particles from both sessions were extracted at full spatial resolution of 1.07 Å pixel size and at 256 pixels box size. Particles corresponding to the two datasets were individually subjected to standard 3D refinement applying solvent-flattened resolution estimation, to CTF refinement (including position-specific beam tilt estimation and per particle defocus estimation) and to Bayesian Polishing trained on 5000 particles for each dataset. Subsequently, particles from both datasets were merged and subjected to standard 3D autorefinement, which produced an overall density at 4.1 Å resolution after post-processing. Next, this consensus refinement was used as a basis for multibody refinement[32] with two segments each representing one spoke of the γ-TuSC (see Supplementary Fig. 2d for details). The resulting unfiltered density segments were merged into a composite reconstruction in UCSF Chimera. Global resolution of the final density was estimated to 3.6 Å after post-processing. All resolution estimates are based on the 'gold standard' Fourier shell correlation (FSC) criterion for independently refined half map reconstructions (FSC = 0.143) within Relion. Local resolution estimates are based on using Relion's local post-processing implementation. The B-factor used during post-processing and local resolution filtering was −300. Data points of all FSC curved are included in the Source Data. ChimeraX[33] was used for visualization of the final cryo-EM density in Fig. 1.

**Model building**. Initial homology models for *C. albicans* γ-tubulin, Spc97 and Spc98 were prepared in Phyre2[34] using human γ-tubulin (PDB 1Z5W) and human GCP4 (PDB 3RIP) as references. The homology models for Spc97, Spc98, and two copies of γ-tubulin were docked into the cryo-EM density as rigid bodies using UCSF Chimera[35] 'Fit in Map'. The rigid body-fitted models were subjected individually to molecular dynamics flexible fitting with one macrocycle of real space refinement using the Namdinator website[36] tool with default parameters. Further refinement and model building were performed in Coot[37]. For the model of γ-tubulin bound to Spc97, residues 36–41, 71–74, and 466–472 were built into the density de novo, and residues 121–127, 203–209, 243–260, 310–319, and 428–437 were deleted, because density was not of sufficient quality. For the model of γ-tubulin bound to Spc98, residues 36–52, 71–74, and 466–471 were built into the density de novo, and residues 121–127, 203–209, 243–260, 310–319, and 428–437 were deleted. For Spc97 and Spc98, sequence homology with the template (human GCP4) was limited and therefore several helices had an incorrect register in the homology models, as obvious from the mismatch between clearly discernible bulky amino acid side chains in the cryo-EM density and in the initial models. For these helices, register was adjusted based on density features and partially using sequence homology with human GCP3 (PDB 6V6B). For the model of Spc97, residues 230–235, 265–272, 503–530, 684–693, 785–802, and 867–871 were built into the density de novo, and residues 110–132, 565–575, and 802–812 were deleted, because density was not of sufficient quality. For the model of Spc98, residues 622–627, 630–638, 647–655, and 776–785 were built into the density de novo, and residues 679–687 and 726–733 were deleted. The Spc97 and Spc98 models are missing an N-terminal region of 91 and 149 residues, respectively, because these regions are not resolved in the cryo-EM density. The initial models for Spc97, Spc98 and the two copies of γ-tubulin were subjected independently to real space refinement in Phenix 1.14[38] with default parameters applying 0.6 restraint weights. After this first round of refinement, the fit between cryo-EM density and atomic models was visually inspected and adjusted in Coot where required. In a last step, the models were combined and refined together into the density as described above.

**Analysis of γ-tubulin conformation and nucleotide occupancy**. All analysis steps detailed below were performed in UCSF Chimera[35]. The overall conformation of γ-tubulins in our cryo-EM density was identified by comparison to X-ray structures of αβ-tubulin dimers in the curved and straight conformations (PDB 4FFB, PDB 5W3F). We focused on the arrangement of α-helices α6 and α7 in β-tubulin, which is the most variable region between the two conformations. β-tubulin

subunits extracted from the X-ray structures were docked as rigid bodies into the two density segments representing γ-tubulin. First, we evaluated the visual overlap between the β-tubulin α-helices and the cryo-EM density. Second, we simulated density segments for the β-tubulin α-helices ('molmap', 3.6 Å resolution) and computed cross-correlation towards the respective segments of the cryo-EM density. Third, we computed R.M.S.D values between the β-tubulin α-helices and the corresponding α-helices in our γ-tubulin models. All three approaches consistently indicate that γ-tubulin is in a straight conformation.

For analysis of γ-tubulin ligand densities, we docked an X-ray structure of GTP-containing β-tubulin (PDB 4FFB) into the two γ-tubulin density segments of our cryo-EM reconstruction. Next, we removed cryo-EM density corresponding to the β-tubulin chains in the X-ray structures ('color zone', 2.5 Å radius). The two remaining density segments in the γ-tubulin regions were overlapping well with the nucleotide ligands in the X-ray structures, but even at low density threshold levels, the γ-phosphate groups of GTP were not covered by the cryo-EM density.

**Analysis and visualization of intermolecular interactions**. Intermolecular interactions were detected using the PISA website tool[39] and visualized in UCSF Chimera. Hydrophobic interactions in the main figures were visualized as follows: For the first interactor, side chains of interfacing residues were shown in atom representation. For the second interactor, the interfacing residues were shown as surface and colored according to hydrophobicity in UCSF Chimera by the 'Render by Attribute-Kd hydrophobicity' function. To visualize the hydrophobic cores stabilizing the helical bundles of the GRIP domain core fold and the GRIP1-GRIP2 domain interface in Supplementary Fig. 3, the Spc α-helices were shown in ribbon representation and amino acid side chains were shown only for the residues contributing to the interaction. Electrostatic interactions were visualized as follows: The interfacing residues for both interactors were shown as surface and colored according to electrostatic properties in UCSF Chimera by the 'Coulombic surface coloring' function.

**Negative stain electron microscopy**. In all, 5 μl of either wild-type γ-TuSC, Spc98$^{\Delta D627-K650}$ mutant γ-TuSC or Tub4$^{\Delta T38-K71}$ mutant γ-TuSC were applied on a 400 copper/palladium mesh grid (PLANO GmbH; Wetzlar-Germany) covered with a carbon layer of ~10 nm thickness according to in-house protocols. The grids were glow discharged for 30 s before the sample was applied and incubated for 30 s on the grid. Grids were blotted on Whatman filter paper 50 and washed via three drops of distilled water and poststained with 3% uranyl acetate. Excessive uranyl acetate was blotted away. Acquisition of datasets was performed on a Talos L120C TEM (Thermo Fisher Scientific) operated at room temperature at 120 kV using the EPU software package (Thermo Fisher Scientific). Particles were imaged with a 4k × 4k Ceta 16 M camera (Thermo Fisher Scientific) at an object pixel size of 0.2552 nm. In total, 997 images of wild-type γ-TuSC, 821 images of Spc98$^{\Delta D627-K650}$ mutant γ-TuSC and 1147 images of Tub4$^{\Delta T38-K71}$ mutant γ-TuSC were acquired. Each of the dataset was acquired in one imaging session.

Image analysis for all datasets was performed in Relion 3.0 Beta. The CTF of micrographs was estimated using gCTF. For all datasets, ~1000 particles were manually selected and extracted at a pixel size of 0.51 nm in boxes of 128 pixels. Particles were subsequently subjected to 2D classification into 50 classes with a translational search range of 20 pixels (2 pixels increment) and a mask diameter of 350–400 Å. 2D classes representing true positive particles were selected and used as references for automated particle picking. The overall number of picked particles was 347,445 for the wild-type γ-TuSC, 154,883 for the Spc98$^{\Delta D627-K650}$ mutant γ-TuSC and 241,205 for the Tub4$^{\Delta T38-K71}$ mutant γ-TuSC. Particles were subjected to 2D classification into 200 classes with parameters as described above. 2D classes representing true positive particles were selected for a second round of 2D classification using the same parameters. Selected classes of the Tub4$^{\Delta T38-K71}$ mutant γ-TuSC were subjected to a third round of classification into 170 classes. Only 2D classes depicting a clearly discernable V-view of the γ-TuSC were used for the analysis. 2D classes that potentially corresponded to side views of the complex or individual spokes were excluded from the analysis, because the overall conformation of the complex could not be discerned from these classes. The final number of particles used for computing the ratios of particles with a straddled appearance were 77,858 for the wild-type γ-TuSC, 11,901 for the Spc98$^{\Delta D627-K650}$ mutant γ-TuSC and 15450 for the Tub4$^{\Delta T38-K71}$ mutant γ-TuSC.

**Multiple sequence alignment**. We compiled sequences for Spc97, Spc98, and γ-tubulin from a broad range of organisms. Sequences were either already annotated in uniprot or identified by Blast search against the respective sequences from *C. albicans*. The following sequences in the format *organism* (Spc97, Spc98, γ-tubulin) were used: *Candida albicans* (Q59PZ2, A0A1D8PS42, O93807), *Saccharomyces cerevisiae* (P38863, P53540, P53378), *Clavispora lusitaniae* (XP_002617032.1, OVF11036.1, C4Y8G4), *Ogataea polymorpha* (XP_018211750.1, XP_018213099.1, A0A1B7SAP4), *Wickerhamomyces ciferrii* (XP_011273399.1, XP_011277192.1, K0KDD8), *Lachancea fermentati* (SCW03932.1, SCW02410.1, A0A1G4M7N1), *Ascoidea rubescens* (XP_020046155.1, XP_020044979.1, A0A1D2VI38), *Schizosaccharomyces pombe* (Q9Y705, Q9USQ2, P25295), *Dictyostelium discoideum* (CAC47949.1, Q95ZG4, Q55AR3), *Coprinopsis cinereal* (XP_002910163.1,

XP_001837207.2, Q7Z9Y2), *Neurospora crassa* (XP_962067.1, XP_960965.1, P53377), *Arabidopsis thaliana* (Q9C5H9, Q9FG37, P38557), *Tetrahymena thermophila* (Q23AE3, Q22ZA9, O00849), *Chlamydomonas reinhardtii* (A8J5J8, A8JBY6 Q39582), *Trichomonas vaginalis* (A2E313, A2E3S1, A2EAH1), *Giardia intestinalis* (A8BD62, A8BFK8, A8BQF3), *Amphimedon queenslandica* (XP_019853680.1, XP_019863806.1, A0A1X7VUX8), *Dendronephthya gigantean* (XP_028391404.1, XP_028405676.1, XP_028392057.1), *Strongylocentrotus_purpuratus* (XP_030831245.1, XP_030837580.1, Q9GYY8), *Ciona intestinalis* (XP_026696722.1, XP_002131421.1, H2XQX6), *Clonorchis sinensis* (GAA57286.1, GAA53692.1, H2KTT6), *Lingula anatine* (XP_013411271.1, XP_013415922.1, XP_013407731.1), *Crassostrea gigas* (XP_011429809.1, XP_011436049.1, K1Q4G7), *Trichinella spiralis* (KRY36071.1, KRY40906.1, A0A0V1BZ12), *Drosophila melanogaster* (Q9XYP7, Q9XYP8, P23257), *Danio rerio* (NP_956416.1, NP_001004513.1, Q7ZVM5), *Podarcis muralis* (XP_028588014.1, XP_028583946.1, XP_028560954.1), *Calypte anna* (XP_030309245.1, XP_030325143.1, A0A091HMN8), and *Homo sapiens* (Q9BSJ2, Q96CW5, P23258). Spc97 and Spc98 sequences were aligned in Promals3D[40]. γ-tubulin sequences were aligned in MATFF[41]. Multiple sequence alignment results were visualized in Jalview[42] using the standard ClustalX colouring scheme and a conservation threshold of 10.

**Sequence-based identification of GCP2-6 subunits**. For analysis of MT system complexity from a phylogenetic perspective (Supplementary Fig. 11 and Supplementary Tables 1 and 2), GCP variants from various organisms were identified based on annotation in the Uniprot database, published literature, PSI-blast E-values against human GCP variants and domain organization (insertions and terminal extensions).

**Analysis of conformation and geometry**. To compare the γ-TuSC conformation in *C. albicans* to the γ-TuSC conformations in the human γ-TuRC (PDB 6V6B) and the *S. cerevisiae* γ-TuSC oligomer in the open (PDB 5FM1) and closed (PDB 5FLZ) states, we split our model of the *C. albicans* γ-TuSC into six segments: GRIP1 and GRIP2 domains of Spc97, GRIP1 and GRIP2 domains of Spc98 and two copies of γ-tubulin. The individual segments were superposed to the respective segments in the human or *S. cerevisiae* γ-TuSC models using the "matchmake" command in UCSF Chimera. The six fitted segments were subsequently combined into a new model and superposed to our model of the *C. albicans* γ-TuSC according to the entire Spc97-γ-tubulin spoke. Conformational variability between the *C. albicans* and human or *S. cerevisiae* γ-TuSC was quantified by measuring the r.m.s.d. values for the six individual segments. To visualize the conformational differences, trajectories linking residues in the *C. albicans* and human or *S. cerevisiae* γ-TuSC were computed and displayed in PYMOL (Pymol v2.1, Schrödinger,).

**Reporting summary**. Further information on research design is available in the Nature Research Reporting Summary linked to this article.

## Data availability

The Cryo-EM density of the γ-TuSC filtered according to local resolution was deposited in the Electron Microscopy Data Bank (EMDB) under accession code EMD-11835. Atomic coordinates for the γ-TuSC were deposited at the Protein Data Bank (PDB) under accession code PDB-7ANZ.

The raw cryo-EM micrograph movie stacks are available from the corresponding authors upon request. Source data are provided with this paper.

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

## Acknowledgements

We thank Tien-Chen Lin (DZNE, Cologne) for helpful discussions. We acknowledge access to the infrastructure of the Cryo-EM Network at the Heidelberg University and support by Götz Hofhaus (Bioquant) and Dirk Flemming (BZH). We acknowledge the services SDS@hd and bwHPC supported by the Ministry of Science, Research and the Arts Baden-Württemberg, as well as the German Research Foundation (INST 35/1314-1 FUGG and INST 35/1134-1 FUGG). This work was supported by the German Research Foundation (DFG; Schi 295/4-4 to E.S.). P.L. received a HBIGS fellowship.

## Author contributions

A.Z. generated the mutant γ-TuSC plasmids, performed wild-type and mutant γ-TuSC expression and purification, and performed the yeast shuffle experiments and phenotype analysis. A.Z. and M.W. performed the anion-exchange chromatography and baculovirus production. A.B. performed functional characterization of the purified γ-TuSC. A.N. acquired negative stain EM data of wild-type and mutant γ-TuSC. E.Z. processed negative stain data. P.L. and A.N. prepared γ-TuSC cryo-EM grids. E.Z. and A.N. acquired cryo-EM data. E.Z. processed cryo-EM data, prepared atomic models, and pursued all related aspects of sequence and structure analysis. All authors discussed and interpreted the γ-TuSC structure. E.S. and S.P. planned and supervised the experiments, analyzed data and together with E.Z. wrote the manuscript.

## Funding

## Competing interests

All authors declare no competing interests.
