## [Peer Review File · Nature Communications]

REVIEWER COMMENTS

Reviewer #1 (Remarks to the Author):

Control of the microtubule cytoskeleton through g-tubulin mediated nucleation is central to the many cellular processes. The structures and organization of g-tubulin containing complexes are thus critical to our understanding of these processes.

Zupa et al describe the cryo-EM structure of the of so-called g-tubulin small complex (g-TuSC) from the fungus *C. albicans* and compare it with other g-tubulin containing structures, including recently solved vertebrate g-tubulin ring complex (g-TuRC) structures. Their reconstruction shows the conformations of sequence inserts found in a subset of fungi, that are located at the interface of the Y-shaped complex. Through protein engineering, the authors show that these inserts are important for complex stability and conformation. They also demonstrate in *S. cerevisiae* that these insertions are important for in vivo function.

The work is expertly performed, and the manuscript is well written and illustrated by mainly clear figures. The discussion is also very interesting on the topic of evolution of g-tubulin-containing microtubule nucleation systems. However, the work overall represents a small advance in our understanding of the mechanisms of microtubule nucleation which are not directly addressed in the current work.

Minor comments:

- Was the purified g-TuSC subjected to mass spec to confirm that the assigned density shown in grey Fig 1a is not attributable to co-purified proteins?
- It would have been useful to have explicitly described the conformation (curved or straight) and nucleotide occupancy of g-tubulin
- Given there are 3 unique proteins in g-TuSC, I could not understand the following text (line167-168): "Collective, insertions in three out of four γ -TuSC subunits form a uniquely extended interaction interface...."
- In Fig 2a and b, the interfaces between Spc97/98 are compared with those of vertebrate GCP2/3 but each interface is depicted using a different scheme, hydrophobic/hydrophilic for Spc97/98 and electrostatic for GCP2/3. Each depiction is logical in isolation given the dominance of these contacts at each interface but direct comparison is impossible without each complex being presented in the same way.

Reviewer #3 (Remarks to the Author):

The manuscript of Zupa and colleagues reports the cryo-EM structure of the g-TuSC from *C. albicans*, and, by comparing with previous g-TuSC structures from vertebrates, discovered that the interactions between spokes have diverged. The authors found that in vertebrates, the interactions between spokes is mainly made of electrostatic interactions, whereas spokes in *C. albicans* are stabilized via a hydrophobic interface. Importantly, the authors show that this unique hydrophobic interface is crucial to maintain a close configuration of the g-TuSC and, by sequence homology in *S. cerevisiae*, found that this interaction is therefore crucial for its function. I found this article very well written and very interesting. The results are solid as well as the interpretations. This work is very complementary to the recently published structures and allows to better understand how g-TuSC functions in lower eukaryotes.

I therefore recommend this manuscript for publication. However, I have some minor points that need to be clarified.

The authors demonstrate in vitro that the insertion D627-K650 of Spc98 stabilizes the complex. To

test its functionality in vivo, the authors use then *S. cerevisiae*. The authors explain this choice because Spc98 "slightly varies in its length between the two species of fungi suggesting a similar function in both organisms". By aligning the sequences, the authors find that the insertion domain would correspond at the region K674-H713 in *C. cerevisiae*. However, even if the insertions are both present in fungi and absent in homo sapiens, the sequences of these insertions do not share a good similarity. Can authors comment on this lack of sequence identity?

If the sequence, even divergent, fulfills the same function, is it possible to replace the Spc98 insertion in *S. cerevisiae* by that of *C. albicans* ? Would it rescue the functionality of the *S. cerevisiae* Spc98? Similarly, is it possible to completely replace the Spc98 of *S. cerevisiae* by that of *C. albicans* ? Would this maintain the function?

The authors also provide an analysis of the sequences in a large number of organisms that allows them to conclude (this is the title of the paragraph), that "the Spc98 insertion evolutionarily compensates for the loss of an extended interaction network in the γ -TuRC". I find this analysis interesting but it is simply a sequence analysis and I think that the title of this paragraph should be tone down, this conclusion remains speculative.

Finally, I am surprised that the work of Brilot & Agard (BiorXiv: <https://www.biorxiv.org/content/10.1101/310813v1>) is not cited in this manuscript, or even discussed. But maybe this work has been published elsewhere and I haven't seen it.

Point-by-Point Reply to Reviewers

We thank the reviewers for their thoughtful and very supportive comments. Below we address the specific points raised by the reviewers and elaborate on the corresponding changes in the manuscript.

Referee #1 (Remarks to the Author):

Control of the microtubule cytoskeleton through γ -tubulin mediated nucleation is central to the many cellular processes. The structures and organization of γ -tubulin containing complexes are thus critical to our understanding of these processes.

Zupa et al describe the cryo-EM structure of the of so-called g-tubulin small complex (γ -TuSC) from the fungus *C. albicans* and compare it with other g-tubulin containing structures, including recently solved vertebrate g-tubulin ring complex (γ -TuRC) structures. Their reconstruction shows the conformations of sequence inserts found in a subset of fungi, that are located at the interface of the Y-shaped complex. Through protein engineering, the authors show that these inserts are important for complex stability and conformation. They also demonstrate in *S. cerevisiae* that these insertions are important for in vivo function.

The work is expertly performed, and the manuscript is well written and illustrated by mainly clear figures. The discussion is also very interesting on the topic of evolution of γ -tubulin-containing microtubule nucleation systems.

We very much appreciate the positive evaluation of our manuscript.

However, the work overall represents a small advance in our understanding of the mechanisms of microtubule nucleation which are not directly addressed in the current work.

We agree with the reviewer that our manuscript does not directly address the actual mechanistic principles of MT nucleation, in the sense of deciphering which oligomeric species of $\alpha\beta$ -tubulin dimers are involved, which are the prevailing $\alpha\beta$ -tubulin dimer contacts during nucleation, what are the dynamics etc.

However, the structural determinants we uncover for the γ -TuSC/CMI system in this work and how they relate to the vertebrate γ -TuRC system indirectly provide important insights into MT nucleation and will be highly relevant for understanding and mechanistic dissection of these processes in the future. In particular, we describe two fundamentally different molecular strategies underlying stabilization of γ -TuSC units in the yeast and human systems and thereby explain the distinct assembly and regulation mechanisms of the microtubule nucleation templates found in these two organisms.

Importantly, the in-depth analysis of γ -tubulin conformations and nucleotide states we now performed in response to a comment of this reviewer and the conclusions that can be drawn from these analyses are important steps towards a more mechanistic understanding of MT nucleation (see below).

Minor comments:

- Was the purified g-TuSC subjected to mass spec to confirm that the assigned density shown in grey Fig 1a is not attributable to co-purified proteins?

We have not pursued mass spectrometry analysis of the purified complexes. However, the SDS-PAGE analysis in Supplementary Fig. 1b indicates that there are no proteins co-purified with the γ -TuSC in a significant stoichiometry. Three major bands are visible on the gel, which can be unambiguously attributed to Spc97, Spc98 and γ -tubulin.

*Furthermore, we have extensively characterized the γ -TuSC from *C. albicans* in a previous publication¹ using tagged versions and deletions of Spc97, Spc98 and γ -tubulin. This work clearly identified Spc97, Spc98 and γ -tubulin as the only constituents of the *C. albicans* γ -TuSC as expressed in insect cells.*

We have now included this into the main text:

“Since we can exclude that proteins were co-purified at significant stoichiometry with the recombinant γ -TuSC (Supplementary Fig. 1b) and because this density loosely connects to the most N-terminal resolved Spc helices, it could represent the missing N-terminal segments of either Spc97, Spc98, or both of the Spc proteins.”

- It would have been useful to have explicitly described the conformation (curved or straight) and nucleotide occupancy of g-tubulin.

We thank the referee for this suggestion. We have conducted a systematic comparison between γ -tubulins in our cryo-EM density and available X-ray structures of $\alpha\beta$ -tubulin dimers in the curved and straight conformations. We have focused our comparison on the two α -helices that deviate most between the two conformations in the $\alpha\beta$ tubulin dimer structures, namely β -tubulin helices six and seven.

First, we docked the X-ray structures as rigid bodies into our cryo-EM density and visually inspected the overlap between our density and the β -tubulin helices in both conformations. This already clearly indicated that our density reflects the straight conformation (new panel d in Fig. 1). Next, we computed cross-correlation values between our cryo-EM density and simulated densities for the β -tubulin helices in both conformations. As expected from the visual inspection, cross-correlation values were significantly higher for the straight conformation, strengthening our assignment of γ -tubulin conformation (new Table 2). Finally, we computed the root mean square deviation (RMSD) of backbone atoms between the two helices in our γ -tubulin model and the β -tubulin models, again observing lower RMSD (thus higher similarity) for the model in the straight conformation (new Table 2).

Next, we systematically analyzed the nucleotide binding pocket of γ -tubulins in our cryo-EM density. We docked an X-ray structure of β -tubulin with GTP as a rigid body and subtracted the cryo-EM density explained by our γ -tubulin model. The remaining density in the nucleotide binding pocket overlaps well with the nucleotide base and ribose in the X-ray structure, but clearly only two of three phosphate groups are covered by the density (new panel e in Fig. 1). This indicates that γ -tubulins in our cryo-EM density are associated with GDP (rather than GTP).

We have included these findings into the main text and added two panels in Fig. 1:

“ γ -tubulin is in a MT nucleation competent conformation

*For $\alpha\beta$ -tubulin dimers, two different conformations have been described. According to the geometry of protofilaments that they occur in, these two conformations have been termed ‘straight’ or ‘curved’². To identify the conformation of γ -tubulins in our cryo-EM reconstruction, we compared them to X-ray structures of $\alpha\beta$ -tubulin dimers in the two different conformations^{3,4} (PDB-4FFB, PDB-5W3H), with a focus on the two α -helices in β -tubulin most variable between the two conformations. Visual inspection of overlap (Fig. 1d), cross-correlation between density segments (Table 2), and root mean square deviation (R.M.S.D) of protein backbone atoms in the atomic models (Table 2) consistently indicate that both copies of γ -tubulin in our cryo-EM reconstruction adopt a ‘straight’ conformation, as typically observed for $\alpha\beta$ -tubulin dimers in stable and ordered microtubules². Thus, γ -tubulins in the fungal γ -TuSC are in a conformation that can promote growth of ‘straight’ microtubule protofilaments without a conformational change⁵. Our cryo-EM density of the *C. albicans* γ -TuSC indicates binding of GDP, rather than GTP (Fig. 1e), and it remains to be addressed whether a nucleotide exchange to GTP is required to promote microtubule nucleation activity as suggested by analysis of γ -tubulin mutants in *S. cerevisiae*⁵.”*

- Given there are 3 unique proteins in g-TuSC, I could not understand the following text (line167-168): “Collective, insertions in three out of four γ -TuSC subunits form a uniquely extended interaction interface....”

We apologize for not having phrased this in an unambiguous manner. It is true that the γ -TuSC contains three unique types of proteins, i.e. Spc97, Spc98 and γ -tubulin, but γ -tubulin is present in two copies, adding up to four individual protein chains / subunits in the complex. The N-terminal insertion in γ -tubulin participates in complex stabilization only for one of the two γ -tubulin copies, i.e. the copy associated with Spc98. To clarify, we now explicitly list the subunits participating in formation of the extended interface:

“Collectively, insertions in three out of four γ -TuSC subunits (namely Spc97, Spc98 and Spc98-associated γ -tubulin) form a uniquely extended interface...”

- In Fig 2a and b, the interfaces between Spc97/98 are compared with those of vertebrate GCP2/3 but each interface is depicted using a different scheme, hydrophobic/hydrophilic for Spc97/98 and electrostatic for GCP2/3. Each depiction is logical in isolation given the dominance of these contacts at each interface but direct comparison is impossible without each complex being presented in the same way.

We fully agree with the reviewer that it is more informative to provide a direct comparison of the interfaces using the same type of scheme, for both hydrophobic and electrostatic interactions. We now included such a comparison in the new Supplementary Fig. 4 and reference it where appropriate in the main text. Due to size limitations and to avoid redundancy in the main figure, we would prefer to keep the main figure as it is, though.

Reviewer #3 (Remarks to the Author):

The manuscript of Zupa and colleagues reports the cryo-EM structure of the g-TuSC from *C. albicans*, and, by comparing with previous g-TuSC structures from vertebrates, discovered that the interactions between spokes have diverged. The authors found that in vertebrates, the

interactions between spokes is mainly made of electrostatic interactions, whereas spokes in *C. albicans* are stabilized via a hydrophobic interface. Importantly, the authors show that this unique hydrophobic interface is crucial to maintain a close configuration of the γ -TuSC and, by sequence homology in *S. cerevisiae*, found that this interaction is therefore crucial for its function. I found this article very well written and very interesting. The results are solid as well as the interpretations. This work is very complementary to the recently published structures and allows to better understand how γ -TuSC functions in lower eukaryotes. I therefore recommend this manuscript for publication.

We appreciate the very positive evaluation of our manuscript.

However, I have some minor points that need to be clarified.

The authors demonstrate in vitro that the insertion D627-K650 of Spc98 stabilizes the complex. To test its functionality in vivo, the authors use then *S. cerevisiae*. The authors explain this choice because Spc98 “slightly varies in its length between the two species of fungi suggesting a similar function in both organisms”. By aligning the sequences, the authors find that the insertion domain would correspond at the region K674-H713 in *S. cerevisiae*. However, even if the insertions are both present in fungi and absent in homo sapiens, the sequences of these insertions do not share a good similarity. Can authors comment on this lack of sequence identity?

Non-peer reviewed work from Brilot and Agard published as a preprint on biorxiv⁶ shows that the Spc98 insertion in S. cerevisiae bridges over towards Spc97 in a similar manner to what we observed in C. albicans, indicating a similar function although this was not specifically mentioned or experimentally addressed in the preprint. We have added this in the main text:

“... slightly varies in its length between the two species of fungi (Supplementary Figs. 5a-c). This suggests a similar function in both organisms which also could be confirmed from a structural perspective in a preliminary cryo-EM study on the S. cerevisiae γ -TuSC⁶.”

Even though the Spc98 insertion has a similar subunit-bridging function in S. cerevisiae and C. albicans, the overall structural architecture of the extended interface has apparently diverged between the two species. In particular, insertions in C. albicans Spc97 and γ -tubulin, which interact with the Spc98 insertion (Figs. 2c, e) and seem to be required for its stabilization (Fig. 3c), are not present in S. cerevisiae (Supplementary Fig. 5). Considering this apparent structural divergence of the extended interface, it is not surprising that also the sequences of the Spc98 insertions have diverged.

If the sequence, even divergent, fulfills the same function, is it possible to replace the Spc98 insertion in *S. cerevisiae* by that of *C. albicans* ? Would it rescue the functionality of the *S. cerevisiae* Spc98? Similarly, is it possible to completely replace the Spc98 of *S. cerevisiae* by that of *C. albicans* ? Would this maintain the function?

Based on the structural divergence of the extended interface described in detail above, neither the isolated Spc98 insertion nor full-length Spc98 protein from C. albicans are expected to complement the respective components in S. cerevisiae. We indeed have confirmed experimentally that full-length C. albicans Spc98 protein cannot functionally replace S.

cerevisiae Spc98 in a complementation experiment. We have included this analysis below in Fig. 1 of the point-by-point reply, but refrained from including it into the manuscript, because the experiment was performed only once and does not contribute to the overall understanding of the *Spc97-Spc98* interaction in *C. albicans*.

Figure 1. *C. albicans* SPC98 cannot complement the function of *S. cerevisiae* SPC98. *C. albicans* SPC98 (cSPC98) and *S. cerevisiae* SPC98 (ScSPC98) both expressed from the *S. cerevisiae* ScSPC98 promoter encoded on the pRS425 plasmid were transformed into *S. cerevisiae* strain *ESM243-3* (*MATa Δspc98::HIS3 pRS316-ScSPC98*). pRS425 was used as control for growth failure in the absence of ScSPC98 function. Serial dilutions of transformants were tested for growth at 30°C on SC-Leu (left) and 5-FOA (right) plates for 2 days. All cells grew equally on SC-Leu plates because of the presence of pRS316-ScSPC98. 5-FOA only allows growth of cells that spontaneously lost the URA3-based plasmid pRS316-ScSPC98. Cells with the empty pRS425 plasmid did not grow on 5-FOA plates, because SPC98 is an essential gene and therefore loss of pRS316-SPC98 is lethal. Cells with the pRS425-ScSPC98 plasmid grew on 5-FOA plates, because the LEU2-based plasmid contained ScSPC98. Cells with the pRS425-cSPC98 plasmid did not grow indicating that *C. albicans* SPC98 cannot complement the function of *S. cerevisiae* SPC98.

The authors also provide an analysis of the sequences in a large number of organisms that allows them to conclude (this is the title of the paragraph), that “the Spc98 insertion evolutionarily compensates for the loss of an extended interaction network in the γ -TuRC”. I find this analysis interesting but it is simply a sequence analysis and I think that the title of this paragraph should be tone down, this conclusion remains speculative.

Together with the high-resolution structural information we present and discuss in this manuscript, our sequence analysis allows us to derive a set of organizational and structural features for the MT nucleation machinery, which all consistently point towards a compensatory role of the Spc98 insertion in evolution. However, we agree with the reviewer that our conclusion eventually remains speculative and therefore we toned down the title. It now reads:

“The Spc98 insertion may have an evolutionarily compensatory role.”

Finally, I am surprised that the work of Brilot & Agard (BiorXiv: <https://www.biorxiv.org/content/10.1101/310813v1>) is not cited in this manuscript, or even discussed. But maybe this work has been published elsewhere and I haven't seen it.

We refrained from directly referencing the work of Brilot and Agard⁶ mentioned by the reviewer for two reasons: First, it still has not been published in a peer-reviewed journal yet, rendering it a preliminary study that might still be subject to significant corrections and changes. Second, the associated structural data (including cryo-EM densities and atomic models) are not publicly available, which precludes any more detailed analysis and comparison with the data presented in our current work.

In the revised version of the manuscript, we nevertheless included the following paragraph in the discussion section, referring to the work of Brilot and Agard:

“Notably, many aspects observed for the γ -TuSC/CMI system in *C. albicans* seem to be generally recapitulated in the related fungus *S. cerevisiae*, as suggested by a preliminary cryo-EM study⁶, but structural data for this study are not publicly available yet, precluding any further in-depth comparison.”

References

- 1 Lin, T. C. *et al.* MOZART1 and gamma-tubulin complex receptors are both required to turn gamma-TuSC into an active microtubule nucleation template. *J Cell Biol* **215**, 823-840, doi:10.1083/jcb.201606092 (2016).
- 2 Brouhard, G. J. & Rice, L. M. Microtubule dynamics: an interplay of biochemistry and mechanics. *Nat Rev Mol Cell Biol* **19**, 451-463, doi:10.1038/s41580-018-0009-y (2018).
- 3 Ayaz, P., Ye, X., Huddleston, P., Brautigam, C. A. & Rice, L. M. A TOG:alphabeta-tubulin complex structure reveals conformation-based mechanisms for a microtubule polymerase. *Science* **337**, 857-860, doi:10.1126/science.1221698 (2012).
- 4 Howes, S. C. *et al.* Structural differences between yeast and mammalian microtubules revealed by cryo-EM. *J Cell Biol* **216**, 2669-2677, doi:10.1083/jcb.201612195 (2017).
- 5 Gombos, L. *et al.* GTP regulates the microtubule nucleation activity of gamma-tubulin. *Nat Cell Biol* **15**, 1317-1327, doi:10.1038/ncb2863 (2013).
- 6 Brilot, A. F. & Agard, D. A. The Atomic Structure of the Microtubule-Nucleating γ -Tubulin Small Complex and its Implications for Regulation. *Preprint at <https://www.biorxiv.org/content/10.1101/310813v1>* (2018).

REVIEWERS' COMMENTS

Reviewer #2 (Remarks to the Author):

The authors have answered my questions, therefore I support this work for publication.